Accuracy of a deep convolutional neural network in detection of retinitis pigmentosa on ultrawide-field images

http://orcid.org/0000-0002-6030-7261 Masumoto Hiroki 1 h.masumoto@tsukazaki-eye.net
Tabuchi Hitoshi 1
Nakakura Shunsuke 1
Ohsugi Hideharu 1
Enno Hiroki 2
Ishitobi Naofumi 1
Ohsugi Eiko 1
http://orcid.org/0000-0002-4813-672X Mitamura Yoshinori 3
1 Department of Ophthalmology, Tsukazaki Hospital , Himeji , Japan
2 Rist Inc. , Tokyo , Japan
3 Department of Ophthalmology, Insutitute of Biomedical Science, Tokushima University Graduate School , Tokushima , Japan
Macknik Stephen
Electronic publication date: 2019 May 7
Publication date: 2019
Volume: 7
Electronic Location ID: e6900
Received 2018 Dec 5; Accepted 2019 Apr 2
Copyright: © 2019 Masumoto et al.
Copyright year: 2019
Copyright holder: Masumoto et al.
License: This is an open access article distributed under the terms of the Creative Commons Attribution License, which permits unrestricted use, distribution, reproduction and adaptation in any medium and for any purpose provided that it is properly attributed. For attribution, the original author(s), title, publication source (PeerJ) and either DOI or URL of the article must be cited.
License URL: https://creativecommons.org/licenses/by/4.0/

Keywords: Neural network, Retinitis pigmentosa, Screening system, Ultrawide-filed pseudocolor imaging, Ultrawide-field autofluorescence

Funding: The authors received no funding for this work.

==============================
Evaluating the discrimination ability of a deep convolution neural network for ultrawide-field pseudocolor imaging and ultrawide-field autofluorescence of retinitis pigmentosa. In total, the 373 ultrawide-field pseudocolor and ultrawide-field autofluorescence images (150, retinitis pigmentosa; 223, normal) obtained from the patients who visited the Department of Ophthalmology, Tsukazaki Hospital were used. Training with a convolutional neural network on these learning data objects was conducted. We examined the K-fold cross validation (K = 5). The mean area under the curve of the ultrawide-field pseudocolor group was 0.998 (95% confidence interval (CI) [0.9953–1.0]) and that of the ultrawide-field autofluorescence group was 1.0 (95% CI [0.9994–1.0]). The sensitivity and specificity of the ultrawide-field pseudocolor group were 99.3% (95% CI [96.3%–100.0%]) and 99.1% (95% CI [96.1%–99.7%]), and those of the ultrawide-field autofluorescence group were 100% (95% CI [97.6%–100%]) and 99.5% (95% CI [96.8%–99.9%]), respectively. Heatmaps were in accordance with the clinician’s observations. Using the proposed deep neural network model, retinitis pigmentosa can be distinguished from healthy eyes with high sensitivity and specificity on ultrawide-field pseudocolor and ultrawide-field autofluorescence images.

Introduction

Retinitis pigmentosa (RP) is a group of hereditary diseases causing progressive visual impairments because of retinal photoreceptor cell degradation (rods and cones) (Fahim, Daiger & Weleber, 2017). Night blindness is a typical clinical feature of the early stage of RP, which is exacerbated by peripheral visual field narrowing and eventually results in loss of central vision (Fahim, Daiger & Weleber, 2017). RP occurs at a frequency of one case per 3,000–7,000 persons with no ethnic preference and is the third leading cause of vision loss in Japan, which experienced a notable increase in the number of RP-related cases of blindness from 1988 to 2008 (Xu et al., 2006). RP is diagnosed based on the patient’s subjective symptoms, fundus findings, visual field examinations, and electroretinogram. There is currently no treatment performed throughout the world for RP that directly acts on the retina (Hartong, Berson & Dryja, 2006). In 2017, voretigene neparvovec-rzyl demonstrated the efficacy in patients with RPE65-mediated inherited retinal dystrophy (Russell et al., 2017) and it was approved by Food and Drug Administration but it was not used worldwide. Induced pluripotent stem cells have been identified as therapeutic tools for treatment of RP (Yoshida et al., 2014); however, study results have remained inconclusive.

There is a concern that exposure to short wavelength light accelerates retinal degeneration, and wearing sunglasses is recommended (Fahim, Daiger & Weleber, 2017). Additionally, services such as vocational training, walking training, and independent life skill training are provided for reduction in vision-related quality of life (Fahim, Daiger & Weleber, 2017). Thus, detecting RP at an early stage is important to prevent worsening in visual function or vision-related quality of life. Furthermore, research on treatments such as artificial retina, gene therapy (Fahim, Daiger & Weleber, 2017), and iPS (Yoshida et al., 2014) has been advanced in recent years.

A striking feature of RP is the appearance of lesions throughout the fundus (Fahim, Daiger & Weleber, 2017). The Optos® camera (Optos Plc, Dunfermline, Scotland) can acquire wide-angle photographs of the fundus that are suitable for diagnosis, because the device can capture images at a 200° range in a non-mydriatic state with a pupil diameter of two mm (Hu, Liu & Paulus, 2016). Ultrawide-field pseudocolor (UWPC) imaging of optoscopic images is versatile and can also be used in diagnosing diabetic retinopathy, vein occlusion, choroidal masses, uveitis, and other similar diseases (Witmer et al., 2013). Black pigment masses are found in the retina in advanced RP. Conversely, it is difficult to diagnose RP sine pigmento or early stage RP in UWPC images (Fahim, Daiger & Weleber, 2017).

Furthermore, the Optos camera can capture fundus autofluorescence (FAF) in the same imaging range which is short wavelength. The FAF represents lipofuscin from predominantly the retinal pigment epithelium (Mercado & Louprasong, 2015). Therefore, FAF is useful for RP diagnosis (Oishi et al., 2016; Robson et al., 2003). In RP eyes, annular enhancement findings are observed in the macular region, whereas granular and mottled attenuation regions are observed in the peripheral region (Oishi et al., 2013). Because RP sine pigmento, which are poorly changed in UWPC images, have functional photoreceptor cell abnormality, it can be considered that there is a change in the ultrawide-field autofluorescence (UWAF) image as well. That is, the UWAF image obtained by FAF is considered to complement the UWPC image.

Efforts to apply image-based diagnostics using machine learning to improve medical care efficiency have been reported (Litjens et al., 2017). Among the current machine learning methods, the deep neural network (DNN) has attracted attention for its use in image-based diagnostics because of its relatively high performance, as compared with conventional machine learning methods (LeCun, Bengio & Hinton, 2015). However, to the best of our knowledge, no study has yet investigated the utility of a DNN model for image-based RP diagnosis. Most recently, several studies regarding the use of a DNN model to assess the UWPC image values for the diagnosis of retinal disorders have been reported by our team (Ohsugi et al., 2017; Nagasawa et al., 2018; Nagasato et al., 2018).

Therefore, the present study aimed to determine whether the DNN in combination with UWPC images is suitable for image-based RP diagnosis. Because it is difficult to diagnose early stage RP using only UWPC images, the classification performance of RP eyes vs normal eyes using DNN with not only UWPC but also UWAF images was examined in this study.

Material and Methods

Data set

The study was approved by the Ethics Committee of Tsukazaki Hospital (Himeji, Japan) (No 171001) and was conducted in accordance with the tenets of the Declaration of Helsinki. An informed consent was obtained from either the subjects of their legal guardians after explanation of the nature and possible consequences of the study (shown in Supplemental Human Studies Consent File 1).

The diagnosis of RP was based on the clinical history, fluorescein angiography, and full-field electroretinograms (ERGs) with the recording protocol conforming to the International Society for Clinical Electrophysiology of Vision standards. The ERGs of all the patients with RP were consistent with rod-cone dystrophy. Patients who were diagnosed with RP had the pathognomonic fundus changes such as attenuated retinal vessels, waxy atrophy of optic nerve head, salt and pepper fundus, and bone-spicule pigment clumping. Patients with uveitis or any disease such as Stargard disease, cone dystrophy that could cause RP-like fundus changes were also excluded. Genetic testing was not performed for each case. As of November 19, 2017, the number of patients with RP registered in the clinical database of the Ophthalmology Department at Tsukazaki Hospital was 226. A total of 72 atypical RP cases such as sector RP and unilateral RP were excluded. A total of 46 cases with complications, such as vitreous hemorrhage, stellate vitreous body, intense cataract distorting the fundus image, previous retinal photocoagulation, and concomitant other fundus diseases, were also exclude. A total of 25 cases in which either UWPC and UWAF images were not obtained were excluded. As a result, the remaining number of patients was 83, and the number of images was 150.

The number of patients with normal eyes registered in the database between October 23, 2017 and November 19, 2017 was 2,926. Among them, the number of patients that a physician checked again for retinal diseases was 594. Furthermore, after excluding patients who did not have UWPC and UWAF taken at the same time, the remaining number of patients was 167, and the number of images was 223.

Totally, 373 UWPC and 373 UWAF images (150, RP; 223, Normal) from the ophthalmology database of Tsukazaki Hospital were used. For each of the UWPC and UWAF images, a model was constructed following the process described below.

In this study, we examined the K-fold cross validation (K = 5). This method has been reported in detail (Mosteller & Tukey, 1968; Kohavi, 1995). All images are divided into K-groups. The right and the left images of the same patients belong to the same group. (K−1) groups are used as training data, and one group is used as validation data. The images of the training data were augmented by adjusting for brightness, gamma correction, histogram equalization, noise addition, and inversion, so that the amount of training data increased by 18-fold. The deep convolutional neural network model was trained with the augmented training data and the validation data and we analyzed the abilities of the deep learning models with the validation data. The training data and the validation data were separated. The process repeated K times until each of the K groups becomes a validation data set shown in Fig. 1.

Figure 1 K-Fold (K = 5) cross validation method.

All images are divided into five groups. Four groups are augmented and then used for training the model, and one group is used as a validation data. The process repeated five times until each of the five groups becomes a validation data. The answers of the neural networks for all images are used for calculating the performance of the neural networks.

Deep learning model

The DNN model called a Visual geometry group—16 (VGG-16) (Simonyan & Zisserman, 2014) used in the present study is shown in Fig. 2. This type of DNN is known to automatically learn local features of images and generate a classification model (Deng et al., 2009; Russakovsky et al., 2015; Lee et al., 2015). All UWPC and UWAF images were resized to 256 × 256 pixels.

Figure 2 Overall architecture of the Visual Geometry Group—16 (VGG-16) model.

VGG-16 comprises five blocks and three fully connected layers. Each block comprises some convolutional layers followed by a max-pooling layer. After flattening the output matrix after block 5, there are two fully connected layers for binary classification. The DNN used ImageNet parameters as the default weights of blocks 1–4 (Nagasato et al., 2018).

Visual geometry group—16 comprises five blocks and three fully connected layers. Each block comprises some convolutional layers followed by a max-pooling layer that decreases position sensitivity and improves generic recognition (Scherer, Müller & Behnke, 2010). The strides of Convolution layers were 1 and the padding of the layers were “same” so the convolution layers only capture the feature of the image, not downsize. The activation function of the layers was ReLU so that we avoid the vanishing gradient problem (Glorot, Bordes & Bengio, 2011). The strides of max pooling layers were 2, so the layers compress the information of the image.

There are a flatten layer and two fully connected layers after the block 5. The flatten layer removes spatial information from the extracted feature vectors, and the fully connected layers compress the information from the previous layers and the last fully connected layers with the activation function, which is softmax, evaluated the probability of each class (in this study 2 classes) and classify the target images. Fine tuning was used to increase the learning speed and to achieve high performance even with less data (Agrawal, Girshick & Malik, 2014). We used parameters from ImageNet: blocks 1 to 4 as default. The weights of convolutional layers and fully connected layers were updated using the optimization momentum stochastic gradient descent algorithm (learning coefficient = 0.0005, inertial term = 0.9) (Qian, 1999; Nesterov, 1983).

The developed prediction model and training were also performed by machine learning with Python Keras (https://keras.io/ja/) using Python TensorFlow (https://www.tensorflow.org/) as a backend. The training and analysis codes are provided as Dataset S1.

Heatmaps

Images were created by overlaying heatmaps of the DNN focus site on the corresponding UWPC and UWAF images. A heatmap of the DNN image focus sites was created and classified using gradient-weighted class activation mapping (Selvaraju et al., 2017). The target layer is as the third convolution layer in block 3. The ReLU (Glorot, Bordes & Bengio, 2011) is represented as backprop_modifier. This process was performed using Python Keras-vis (https://raghakot.github.io/keras-vis/).

Outcomes

Area under the curve (AUC), sensitivity, and specificity were determined for UWPC and UWAF images using the DNN model described above.

We created five models and five receiver operating characteristic (ROC) curves in five times of the process. Images judged to exceed a threshold were defined as positive for RP, and a ROC curve was created.

For sensitivity and specificity, the optimal cutoff values, which are the points closest to the point at which both sensitivity and specificity are 100% in each ROC curve, were used (Akobeng, 2007). Derivation of the ROC curve was performed using Python scikit-learn (http://scikit-learn.org/stable/tutorial/index.html).

Statistical analysis

For comparing patient background, age was tested using Student’s t-test. Fisher’s exact test was performed for comparing the male–female and right-and-left ratios. In all cases, P < 0.05 was considered statistically significant. These statistical processes were performed using Python Scipy (https://www.scipy.org/) and Python Statsmodels (http://www.statsmodels.org/stable/index.html).

For AUC, a 95% confidential interval was obtained by assuming a normal distribution and using the average and standard deviation of five ROC curves. The 95% confidence intervals (CI) of sensitivity and specificity were calculated assuming a binomial distribution.

Derivation of the CIs of AUC, sensitivity and specificity were obtained using Scipy.

Results

Background

The patient background is described in Table 1. There were no significant differences in age, sex, or the ratio between right and left eyes between the normal and RP groups.

Table 1 Background characteristics of study participants.

	Normal	RP	P-value	
N	223	150		
Age	64.0 ± 14.0 (11–78)	61.1 ± 15.1 (19–87)	P = 0.06 (Student’s t-test)	
Sex, female	123 (55.2%)	74 (49.3%)	P = 0.29 (Fisher’s exact test)	
Eye, left	119 (53.4%)	70 (46.7%)	P = 0.21 (Fisher’s exact test)	
Notes:

There are no significant differences in age, female ratio and left ratio between normal images and retinitis pigmentosa images.

Age (years) is reported as the mean ± standard deviation with (range).

Sex, eye are shown as number with (%).

RP, Retinitis Pigmentosa.

Evaluation of model performance

The average AUC of the UWPC and UWAF groups were 0.998% (95% CI [0.995–1.0]) and 1.0% (95% CI [0.999–1.0]), respectively. The one example of the ROC curves of the UWPC and UWAF groups is shown in Fig. 3. The sensitivity and specificity of the UWPC group were 99.3% (95% CI [96.3%–100%]) and 99.1% (95% CI [96.1%–99.7%]), and those of the UWAF group were 100% (95% CI [97.6%–100%]) and 99.5% (95% CI [96.8%–99.9%]), respectively. As for the sensitivity and specificity, there are no significant difference between the UWPC and UWAF groups (sensitivity: P = 0.22, specificity: P = 0.64)

Figure 3 Receiver operating characteristic (ROC) curve of retinitis pigmentosa (RP).

A example of ROC curve of the UWPC and the UWAF.

When the threshold was set to 0.5, one RP image was misclassified as Normal for UWAF, and two Normal images were misclassified as RP and one RP image was misclassified as Normal for UWPC.

Heatmap

Images were created by overlaying heatmaps of the focus site of the DNN on the corresponding UWPC and UWAF images. An example is presented in Fig. 4. In both UWPC and UWAF images, points of interest on the heatmaps accumulated in all of the classic ophthalmoscopic triad of RP which consists of bone spicule pigmentation, waxy pallor of the optic disc, retinal vessel attenuation. This finding suggests that the proposed DNN model may identify RP by paying attention to suspected lesion sites.

Figure 4 The images and their heatmaps of (A) ultrawide-field pseudocolor (UWPC). (B) Ultrawide-field autofluorescence (UWAF).

In both UWPC and UWAF images, points of interest on the heatmaps accumulates in the bone spicule pigmentation of the fundus, which is characteristic of retinitis pigmentosa.

Discussion

The results of the present study indicated that the proposed DNN model could sufficiently distinguish RP from a normal fundus with high sensitivity and specificity (UWPC: sensitivity = 99.3%, specificity = 99.1%; UWAF: sensitivity = 100%, specificity = 99.5%) with both UWPC and UWAF images. In this way, the multilayered DNN model presents the advantage of constructing an optimum structure for learning and of identifying local features of complex images with subtle individual differences (Deng et al., 2009; Russakovsky et al., 2015; Lee et al., 2015).

Because UWAF images can sometimes reveal the findings that are not clearly shown on UWPC images, sensitivity was expected to be higher for UWAF images than UWPC images; however, there was no significant difference between them. The reason for this is because both sensitivity and specificity in UWPC images are close to 100%, the results that would exceed the UWPC sensitivity and specificity, which constitutes a significant difference, cannot be obtained.

For both UWPC and UWAF images respectively, only one RP image was misclassified as Normal, and these two misclassified images were from different RP cases. Since the misclassified images in the UWAF and UWPC images were derived from different cases, both images should be classified by neural network to prevent overlooking.

Simultaneously obtaining UWPC and UWAF images at using the Optos camera, which is a non-mydriatic examination, without a significant increase in time is relatively easy. Even in areas without an ophthalmologist, if the UWPC and UWAF images can be obtained, it is possible to acquire necessary image data to distantly diagnose RP.

However, since RP is a very rare disease with a prevalence of less than 0.1%, to identify only one RP patient, at least 3,000 healthy persons would have to be examined (Fahim, Daiger & Weleber, 2017). A scheme in which the ophthalmologist diagnose all acquired images is very difficult. Using the DNN model, early detection, and screening of RP can be more efficiently performed.

As the first limitation of the present study, we aimed to evaluate the ability of the combination of the DNN model with UWPC or UWAF images to distinguish between normal and RP eyes, rather than distinguish RP from other fundus diseases or coexisting complications. We focus only on the difference between the RP eyes and the normal eyes. The second limitation is that fundus images of patients with vitreous hemorrhage and advanced cataract were excluded from this study. This identification system presumes screening in telemedicine. There are several complications related to RP; thus, it is impossible to distinguish between these entities and RP from fundus photographs alone (Fahim, Daiger & Weleber, 2017). Confirmation of diagnosis requires conventional diagnostic techniques performed by an ophthalmologist, such as a mydriatic eye fundus examination, retinal electrography, and Goldmann perimetry (Fahim, Daiger & Weleber, 2017). In the future, it is necessary to accumulate more Optos image data of RP to improve the reliability of the discrimination ability of the proposed model.

The lesion of RP is outside of the vessel arcade and the Optos is necessary for the diagnosis of RP. There are no artificial intelligences for diagnose not only for RP but also for other diseases, although our team report several studies about the DNN model for the diagnosis of retinal detachment, glaucoma, diabetic retinopathy, etc. We have to do further experiments in which we classify RP images with non-RP not normal images and construct the DNN model which can diagnose RP regardless of other diseases such as retinal detachment.

Conclusion

In conclusion, the proposed method was found to be highly sensitive and specific with both UWPC and UWAF images for the differential diagnosis of RP. It was suggested that our method has a possibility of detecting RP in clinical practice. However, further experiments are required to develop the artificial intelligence for diagnosis of RP.

Supplemental Information

Supplemental Information 1 Python code.

Raw data of the python code for training and analysis.

Click here for additional data file.

The authors would like to thank Enago and Cosley Nakaba for the English language review. We wish to thank Masayuki Miki and Shoji Morita (Department of Ophthalmology, Tsukazaki Hospital, Himeji, Japan).

Additional Information and Declarations

Competing Interests

Author Contributions

Human Ethics

Data Availability

Hiroki Enno is an employee of Rist Inc.

Hiroki Masumoto conceived and designed the experiments, performed the experiments, analyzed the data, contributed reagents/materials/analysis tools, prepared figures and/or tables, authored or reviewed drafts of the paper, approved the final draft.

Hitoshi Tabuchi conceived and designed the experiments, prepared figures and/or tables, authored or reviewed drafts of the paper, approved the final draft.

Shunsuke Nakakura authored or reviewed drafts of the paper, approved the final draft.

Hideharu Ohsugi authored or reviewed drafts of the paper.

Hiroki Enno contributed reagents/materials/analysis tools, authored or reviewed drafts of the paper.

Naofumi Ishitobi performed the experiments, contributed reagents/materials/analysis tools, authored or reviewed drafts of the paper.

Eiko Ohsugi authored or reviewed drafts of the paper, approved the final draft.

Yoshinori Mitamura authored or reviewed drafts of the paper, approved the final draft.

The following information was supplied relating to ethical approvals (i.e., approving body and any reference numbers):

The study was approved by the Ethics Committee of Tsukazaki Hospital (Himeji, Japan) (No 171001).

The following information was supplied regarding data availability:

The raw images are available at:

Kameoka, Masahiro (2018): masumoto RP data RP FAF. figshare. Fileset. DOI 10.6084/m9.figshare.7397486.v2.

Kameoka, Masahiro (2018): masumoto RP data normal FAF. figshare. Fileset. DOI 10.6084/m9.figshare.7397495.v1.

Kameoka, Masahiro (2018): masumoto RP data normal optos. figshare. Fileset. DOI 10.6084/m9.figshare.7403825.v1.

Kameoka, Masahiro (2018): masumoto RP data RP optos. figshare. Fileset. DOI 10.6084/m9.figshare.7403831.v1.

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
