# Peer review of "Accuracy of a deep convolutional neural network in detection of retinitis pigmentosa on ultrawide-field images"

_PeerJ, doi:10.7717/peerj.6900_

## Round 0.1 · original submission · Major Revisions

Your manuscript has been seen by three reviewers and based on their comments you should make major revisions before resubmitting for re-review.

·

Basic reporting

RAW data is not provided. Please consider using some sharing services. For example, Figshare is an accademic service and should be free for only 746 original images.
No other comments.

Experimental design

Very old people (>90y) also have RP like fundus. Were they excluded?
No other comments.

Validity of the findings

Authors evaluated deep convolutional neural networks in detection of retinitis pigmentosa on ultrawide-field images. Despite of relatively small size data, the accuracy was very high. However, the real world problem is that the patients with early stage of retinitis pigmentosa (RP) are often overlooked. As the authors said, false positive is OK, but false negative is not good in screening. If the authors present some cases with early RP and their accuracy, the readers will agree to the authors' DNN model is useful for detecting the RP.
No other comments.

Additional comments

In the acknowledgements section, Masyuki may be Masayuki.

Reviewer 2 ·

Basic reporting

The level of English is below acceptable standards. Composition of sentences is often weird, e.g., second sentence in abstract: “A single-facility, cross-sectional study.” A single-facility study of what/where/who?
In abstract, I don’t think that “patients…. were used”. I suggest that observations and measurements were used to build the model, not parents themselves.
“Training with a convolutional neural network on these learning data objects was conducted and a deep learning model was constructed.” - Convolutional networks technique is a deep learning method by definition. No need to mention it twice.
I don’t think there is a need to include abbreviations to the abstract.
Authors do not refer to supplementary materials in their manuscript (this should be fixed). However, there were some supplementary files and I am not sure why authors attached a copy the manuscript formatted for Journal of Ophthalmology.
It would be helpful to include a database of images as a raw supplementary data.

Experimental design

Description of methods in not sufficient:
It would be useful to describe convolutional network in more details (What was the activation function? What was the stride?, etc.).
Outcomes is a strange subsection that contains only one sentence. This section should be expanded (e.g., by adding basic definition and formulas).

Validity of the findings

Results section: “There were no significant differences in age….” – such claims must be supported statistically.
Discussion is not comprehensive and does not contain any sort of interpretation.
The manuscript lacks “Conclusion” section.

Reviewer 3 ·

Basic reporting

There are some terms that seem mistranslated
Eg pyramidal cells instead of cones.
I think the manuscript would benefit from a review by an English speaking Japanese ophthalmologist to check there are no other errors like this. Some specific examples are listed below.

In abstract Line 20
“Totally, 373 UWPC…” would be better worded “In total, 373 UWPC…”
In introduction
Line 34 “pyramidal cells” should be “cones”
Line 36 “center” should be “central”
Line 42 It is not clear what “which is spread globally” means
Is it the treatment or the disease spreads globally (throughout the retina)
or the treatment being spread around the world?
Is it the globe of the eye or the globe of the world? (internationally)
Line 91 “sesame salt-like” I think should read as “salt and pepper”

Experimental design

The authors present on the ability of AI/Deep Neural Networks to detect retinitis pigmentosa.
We have seen some major recent papers describing the ability of AI to diagnose common diseases such as diabetic retinopathy, glaucoma and age-related macular degeneration but there is little data in the ability of AI to detect and correctly diagnose rarer retinal diseases.
The authors have a large series of retinal images from a clinic of RP patients.

There is no clear gold standard for RP
Electrophysiology and genetics could be compared to imaging but this is beyond the scope of this research now but will be interesting to see in coming years whether AI can detect early RP

RP is a wide range of diagnoses. A more broad term of inherited retinal diseases would include some disorders with very characteristic and distinct images.
The authors should be more specific about what disorders were excluded from the analysis, eg Stargardts disease, cone dystrophy etc.

Validity of the findings

It is not clear the numbers that were used in the teaching versus the validation part of the experiment.
I think this could be added to table 1

The paragraph beginning with line 209 is not really related to the results of the underlying research presented and might be better in the introduction.

I think a bettter discussion will be on how recognition of RP can be added to other AI being used for other retinal disease so that RP which is rare doesn't get missed or misdiagnosed.

Additional comments

Does the heat map highlight any of the classic ophthalmoscopic triad of RP?
Bone spicule pigmentation, Waxy pallor of the optic disc, Retinal vessel attenuation.
Surprisingly there is little data on the sensitivity and specificity of these signs in RP from the clinical literature.

---

## Round 0.2 · Minor Revisions

I am happy to say that the reviewers feel that your revised manuscript needs only minor revisions. Please address each and every one of these suggestions in your revision. I am optimistic that the manuscript will be acceptable, if so.

·

Basic reporting

Authors evaluated deep convolutional neural networks in detection of retinitis pigmentosa on ultrawide-field images. Despite of relatively small size data, the accuracy was very high.

Experimental design

Authors should declare that the right and the left images of same patient belong to same group.

Validity of the findings

The uploaded images are useful. Some of images, for example [email protected], is not like RP. If the ERGs were taken precisely, the diagnosis would be true.

Additional comments

In this revised version, the manuscript has been well improved according to the editor and reviewers' comments.

Reviewer 2 ·

Basic reporting

Authors have address the majority of my comments. Additional suggestions are in the section "General comments for the author".

Experimental design

Authors have address the majority of my comments. Additional suggestions are in the section "General comments for the author".

Validity of the findings

Authors have address the majority of my comments. Additional suggestions are in the section "General comments for the author".

Additional comments

Authors stated that "I already upload the raw images to Figshare." Will these figure be available for readers?
* * *
/// The manuscript lacks “Conclusion” section. ///
Reply:I am sorry but I can’t what you mean. Our manuscript surely contains the conclusion section.

The so-called "conclusion" contains only one sentence. It is barely a conclusion. Authors must expand this section.

---

## Round 0.3 · accepted · Accept

Thank you for your revisions. I am happy to recommend this manuscript for publication.

#